# Electrochemical Impedance Spectroscopy for the Sensing of the Kinetic Parameters of Engineered Enzymes

**DOI:** 10.3390/s24082643

**Published:** 2024-04-20

**Authors:** Adriána Dusíková, Timea Baranová, Ján Krahulec, Olívia Dakošová, Ján Híveš, Monika Naumowicz, Miroslav Gál

**Affiliations:** 1Department of Molecular Biology, Faculty of Natural Sciences, Commenius University, Ilkovičova 6, 842 15 Bratislava, Slovakia; dusikova7@uniba.sk (A.D.); jan.krahulec@uniba.sk (J.K.); 2Department of Inorganic Technology, Faculty of Chemical and Food Technology STU in Bratislava, Radlinského 9, 812 37 Bratislava, Slovakia; timea.baranova@stuba.sk (T.B.); olivia.dakosova@stuba.sk (O.D.); jan.hives@stuba.sk (J.H.); 3Faculty of Chemistry, University of Białystok, ul. K. Ciołkowskiego 1K, 15-245 Białystok, Poland; monikan@uwb.edu.pl

**Keywords:** engineered enterokinase, enzyme, impedance spectroscopy, kinetic parameters

## Abstract

The study presents a promising approach to enzymatic kinetics using Electrochemical Impedance Spectroscopy (EIS) to assess fundamental parameters of modified enteropeptidases. Traditional methods for determining these parameters, while effective, often lack versatility and convenience, especially under varying environmental conditions. The use of EIS provides a novel approach that overcomes these limitations. The enteropeptidase underwent genetic modification through the introduction of single amino acid modifications to assess their effect on enzyme kinetics. However, according to the one-sample *t*-test results, the difference between the engineered enzymes and hEKL was not statistically significant by conventional criteria. The kinetic parameters were analyzed using fluorescence spectroscopy and EIS, which was found to be an effective tool for the real-time measurement of enzyme kinetics. The results obtained through EIS were not significantly different from those obtained through traditional fluorescence spectroscopy methods (*p* value >> 0.05). The study validates the use of EIS for measuring enzyme kinetics and provides insight into the effects of specific amino acid changes on enteropeptidase function. These findings have potential applications in biotechnology and biochemical research, suggesting a new method for rapidly assessing enzymatic activity.

## 1. Introduction

Enteropeptidases are key enzymes in the human digestive process, responsible for the activation of digestive enzyme precursors such as trypsinogen to trypsin, which in turn activates other pancreatic zymogens critical for nutrient absorption [1]. A detailed kinetic analysis of these enzymes, particularly when genetically modified to enhance their properties, is vital for both understanding their biological functions and their application in therapeutic and industrial contexts [2]. Despite advances in enzymatic studies, accurately determining the kinetic parameters of modified enteropeptidases remains a challenge, impeding the full exploitation of their potential [3]. According to the MEROPS classification, enterokinase (enteropeptidase) EC 3.4.21.9 falls under the clan of PA proteases, subclan PA(S), family Appendix A, and subfamily Appendix A peptidases. Enterokinases’ structural features demonstrate extraordinarily high similarity even in different species. For example, in humans, an enterokinase gene has been localized on chromosome 21q21 [4]. Human and bovine nucleotide sequences present 85% identity in their coding regions, while the encoded amino acid sequences are 82% identical [4,5]. It is composed of two subunits: a heavy chain (structural subunit) and a light chain (catalytic subunit). Both subunits are linked by a single disulfide bond [6]. Human enterokinase light chains cleave internal peptide and protein bonds at sites that are highly conservative in the cleavage site of bovine trypsinogen: DDDDK↓I [7]. Even though, because of their high specificity, enterokinases are used for removing tags from recombinant proteins, their disadvantage is also, although to a much lesser extent, unwanted proteolysis events at non-specific sites. This problem can be solved on the level of either enzyme or recognition sequence. For example, substituting lysine for arginine in the cleavage site mostly results in a lower concentration of enteropeptidases needed for cleavage, but it does not affect the undesired cleavage in other sequences [8]. In our study, we attempted to introduce several mutations on the enterokinase light chain catalytic domain to improve the specificity activity features.

Traditional methods for enzyme kinetics, such as fluorescence spectroscopy, or other, usually indirect experimental methods, face limitations regarding environmental versatility, cost, and operational complexity. These limitations are particularly pronounced when analyzing samples with varying optical properties or in complex matrices, restricting the comprehensive assessment and utilization of engineered enzymes [3,9,10,11,12]

Innovative methodologies are necessary to determine critical kinetics parameters, such as the maximum reaction rate (*v*_max_), Michaelis constant (K_m_), and turnover number (k_cat_), due to discrepancies in their values [6,13,14,15,16,17,18]. This study aimed to address the aforementioned limitations by utilizing electrochemical impedance spectroscopy (EIS) to determine the kinetic parameters of genetically engineered enteropeptidases. The objective of this study was to validate EIS as a versatile, cost-effective, and user-friendly alternative with broad applicability across diverse experimental conditions. EIS is recognized for its sensitivity and precision in biological applications and offers new insights into enzyme kinetic analysis [19,20,21,22,23,24]. Moreover, EIS provides comparable speed and accuracy to traditional fluorescence spectroscopy, while also reducing operational costs and increasing user-friendliness. This makes EIS a significant advancement in enzymatic studies [19,20,23].

The research started with genetically engineering enteropeptidases to enhance enzymatic performance by introducing specific amino acid alterations. The modified enzymes were then analyzed using traditional fluorescence spectroscopy and EIS to evaluate the efficacy and reliability of the EIS method in determining kinetic parameters. This study not only examined the impact of genetic modifications on enteropeptidase activity but also assessed the practicality and accuracy of EIS compared to the established method. Moreover, this study aimed to bridge the gap between traditional enzymatic data and electrochemical insights [14,19,23]. This approach promotes integration in enzyme kinetics and helps establish optimal reaction conditions.

By aligning electrochemical techniques with the complex biochemistry of enteropeptidases, previous findings were validated, and novel therapeutic applications are made possible. This dual contribution highlights the usefulness of interdisciplinary approaches in resolving longstanding biochemical ambiguities and improving therapeutic enzyme applications.

## 2. Materials and Methods

### 2.1. Impedance Measurements

Electrochemical measurements were performed using a potentiostat PGSTAT 302N (Metrohm A.G., Herisau, Switzerland). Impedance data were collected in the range from 10 kHz to 10 Hz. Each impedance curve consisted of at least of 60 measured points. Electrochemical data from EIS measurements were analyzed using NOVA 2.1 software (Metrohm A.G., Herisau, Switzerland). A screen-printed electrode (SPE) system was used where the working electrode (WE) and counter electrode (CE) were boron-doped diamond electrodes. The reference electrode (RE) was Ag|AgCl|3M KCl separated from the SPE system. The total volume of the measured solution was 500 μL. The concentration of all mutant forms of hEKL used during the experiments was 2.5 nM. All the measurements were carried out in the solution consisting of 10 mM CaCl_2_ (p.a., Ubichem, Budapest, Hungary), 10% DMSO (p.a., Sigma-Aldrich, Steinheim, Germany), and 25 mM TRIS (p.a., Serva, Heidelberg, Germany) at 37 ± 0.1 °C using Mini Dry Bath BSH200-HL (Benchmark Scientific, Inc., Sayreville, NJ, USA).

### 2.2. Mutations of hEKL

Mutations in hEKL were introduced using a series of fusion PCRs that targeted the activity-catalyzing centers. We introduced mutations K99R, N101S, N95E, N95D, and D100N, and we marked them Mut0, Mut1, Mut2, Mut3, and Mut4, respectively (Appendix A). Using the restriction endonucleases *NotI* and *XhoI*, we cloned the individual mutated forms into the pGAPZ*α*C vector. The resultant plasmids were named pGAP-mut0 hEKL, pGAP-mut1 hEKL, pGAP-mut2 hEKL, pGAP-mut3 hEKL, and pGAP-mut4 hEKL. These were specifically integrated into the chromosome of *P. pastoris* Y11430 at the promoter of the glyceraldehyde 3-phosphate dehydrogenase gene. Prior to the transformation, the expression plasmids were linearized by *Avr*II restriction endonucleases to ensure the site-specific integration in the mentioned region. Subsequently, the overproduction of the mutated forms of hEKL was performed in a small-scale bioreactor and tested after purification by IMAC and ion-exchange chromatography.

### 2.3. SDS-PAGE and Testing of Activity on Trx-DCD1 Substrate

The method in Figure 1 was SDS-PAGE, which was performed according to Smith, B. J. [25]. The cleavage of the Trx-DCD1 was performed according to Pepeliaev, S., et al. [26]. Briefly, the technique relied on the enzyme’s ability to identify and split the specific amino acid sequence DDDDK within the thioredoxin-dermcidin (Trx-DCD1) fusion protein molecule. Trx-DCD1 was produced in *E. coli* and thoroughly purified using IMAC and ion-exchange chromatography. In the experiment, samples of the hEKL were diluted from 4 to 100 times in a 10 mM Tris-HCl solution with a pH of 8. Then, 1 µL of the diluted enzyme solution was combined with 4 µL of Trx-DCD1 with a concentration of 3.68 mg/mL, and the mixture was left to incubate for 1 h at 37 °C. Afterward, the mixture was treated with 20 µL of sample buffer to denature it before being loaded onto a 16% SDS-PAGE gel. A positive control was included, using enterokinases from Invitrogen (EKMax^®^, Thermo Fisher Scientific, Waltham, MA, USA), which had a specific activity of 2000 U/mg of protein.

## 3. Results and Discussion

### 3.1. Preparation of Mutant Forms of the Enzyme

Mutations in this work were selected based on amino acid alterations in the sequences of higher organisms with previously studied sequences of the gene hEKL. We selected these mutations based on high enzyme–substrate affinity or catalytic efficiency. The mutated forms were named Mut0, Mut1, Mut2, Mut3, and Mut4. The strain Mut0 was specified by its substitution of lysine to arginine in position 99. The strain Mut1 had a substitution of asparagine in position 101 to serine, because serine in pork enterokinases is placed exactly in this position. The mutation had a substitution of Mut2 at position 95, where asparagine was exchanged for glutamic acid, whereas this exchange was made due to the glutamic acid at this position in mouse and rat enterokinases. Mut4 was modified to replace asparagine with aspartic acid at position 100. This exchange followed a previous Mut2 exchange, with a change to a different amino acid with a negative charge. Mut3 was modified by replacing aspartic acid at position 95 with asparagine. This exchange resulted from bovine enterokinases, which contain asparagine at this position (Table 1 and Figure 1).

### 3.2. Characterization of Basic Kinetic Parameters of Mutant Enzymes

#### 3.2.1. Fluorescence Measurements

UV-Vis and fluorescence spectroscopy are commonly used to measure the biochemical kinetic properties of enzymes, including *v*_max_, K_m_, and k_cat_. However, for certain substrates, such as some fusion proteins, these methods may not yield distinct spectra. In such cases, electrochemical impedance spectroscopy can be beneficial for enzyme characterization.

Fluorescence spectroscopy was initially used to establish the baseline data. Then, the calibration curve was constructed and the relationship between fluorescence intensities at 420 nm over time at various substrate concentrations was studied. The plot of 1/*v*_init_ against 1/[S], where *v*_init_ was the initial velocity of β-naphthylamine production and [S] was the concentration of the GD4K-NA substrate, allowed for the determination of the kinetic parameters for the respective engineered enzymes. The *v*_max_ was determined by taking the reciprocal of the *y*-axis intercept of the linear plot, and the Michaelis constant (K_m_) was found by taking the reciprocal of the *x*-axis intercept. The turnover number, k_cat_, was calculated using the formula k_cat_ = *v*_max_/[E], with [E] representing the concentration of the enzyme. The kinetic parameters of the different engineered forms of enteropeptidases obtained by fluorescence spectroscopy are summarized in Table 2.

#### 3.2.2. EIS Measurements

UV-Vis or fluorescence spectroscopy cannot determine the kinetic parameters of enzymatic reactions in turbid, colored environments or when fusion proteins are involved. In such cases, an alternative experimental method is necessary to determine enzyme kinetic parameters. Electrochemical impedance spectroscopy appears to be such a method.

Figure 2 shows a typical impedance curve with a line indicating how to determine the electrolyte resistance parameter for a given enzyme and substrate concentration. All such curves were measured at a potential corresponding to the open-circuit potential (OCP), which was determined prior to each experiment. For this potential, no electrochemical reaction was expected, and pure capacitive behavior was observed. As a BDD electrode was used, which did not have a perfectly smooth surface, it was necessary to use a constant phase element (CPE) instead of a pure capacitor when fitting the EIS measurement results.

The impedance measurements yield several key parameters. The conductivity of the solution and the electrode’s capacitance may change after proteolytic cleavage, which can be used to evaluate the biochemical characteristics of human enteropeptidases. It is important to assess how the solution’s resistance and/or the electrode’s capacitance change over time post-cleavage. Our findings, illustrated in Figure 2 and Figure 3, showed a reduction in the resistivity of the solution when EKs were added to the GD4K-NA substrate mixture.

The cleavage of the substrate by enteropeptidases resulted in the formation of two species within the solution that had higher conductivity. Therefore, our electrochemical analysis focused primarily on the solution’s resistance (*R*s). We determined the enzyme’s kinetic parameters by observing how the solution’s resistance changed over time after introducing EKs to the substrate solution. The initial reaction rates were calculated by analyzing the changes in resistance across different substrate concentrations, similar to the analysis of changes in fluorescence intensity over time.

The Michaelis–Menten plot was drawn from impedance measurements, as shown in Figure 4.

The biochemical parameters of the enteropeptidase, including the Michaelis constant K_m_, the turnover number k_cat_, and their ratio, were determined by analyzing the linear relationship between the reciprocal of initial velocities and the reciprocal of substrate concentrations. Table 3 presents a comprehensive summary of these comparisons.

These data align closely with those obtained through fluorescence spectroscopy. According to the paired *t*-test, the two-tailed *p* value was higher than 0.05 (0.8678), which meant that by conventional criteria, this difference between fluorescence and EIS data was considered to be not statistically significant. This meant that the EIS technique could be used instead of traditional fluorescence measurements to determine the basic kinetic parameters of enzymes. Furthermore, EIS could be used in experimental conditions where fluorescence spectroscopy may not be effective, such as in turbid or colored environments, or in the case of fusion proteins where cleavage by enterokinases does not provide a sufficiently pronounced fluorescence signal. Therefore, our experiments confirmed the suitability of EIS for kinetic studies on enzymatic reactions [19,20,21]. To evaluate the accuracy of the EIS method in determining the kinetic parameters of hEKL, we compared our findings with those previously reported in the literature [6,8,14,17,18,26].

Enzyme molecules may aggregate or undergo conformational changes that affect the availability of active sites at higher concentrations, thus impacting the turnover number. It is probable that this phenomenon is an artifact of the experimental conditions rather than a genuine variation in the enzyme’s intrinsic catalytic ability. Furthermore, at higher enzyme concentrations, enzymes may interact with molecules other than the substrate, including each other, which could result in inhibition (competitive, non-competitive, or allosteric) and impact their activity, ultimately affecting the observed turnover number. Achieving absolute substrate saturation can be difficult in real-world experiments, especially for enzymes with a high affinity for their substrates (low K_m_). To avoid confusion, it is important to ensure that the system reaches the ideal of substrate saturation. It is crucial to maintain a consistent environment to accurately measure enzyme activity. Variations in pH, temperature, and ionic strength can affect enzyme activity, which can lead to situations where the turnover number appears to depend on enzymatic concentration. For instance, if more enzymes are added, it may indirectly affect the enzyme’s performance by altering the buffer’s pH or ionic strength. Table 4 shows the kinetic parameters of the selected engineered enzymes using higher enzyme concentrations.

Table 4 presents an analysis of the basic kinetic parameters, including the Michaelis–Menten constant (K_m_) and the turnover number (k_cat_), for the selected genetically engineered enzyme variants. The data show that the kinetic parameters remained consistent within the range of concentrations tested, indicating that increases in enzyme concentration did not significantly affect these critical biochemical attributes. This observation highlights the enzyme’s stable catalytic properties under varying experimental conditions, which is crucial for reliable biochemical analysis and application.

The results presented in Table 2, Table 3 and Table 4 are consistent with established findings in the scholarly literature. The modifications introduced and methods employed in our study were validated by this alignment, supporting our hypothesis and the efficacy of our engineered enzymes in mimicking natural kinetic behavior under assay conditions.

Additionally, the parameters derived from EIS measurements showed a high degree of concordance with those obtained through conventional fluorescence spectroscopic techniques, as shown in Figure 5. This agreement validated EIS as an alternative analytical tool for enzymatic studies and underscored its potential utility in environments where traditional fluorescence methods may encounter limitations. The agreement between EIS and conventional methods confirmed the reliability and precision of electrochemical approaches in capturing the nuanced dynamics of enzyme–substrate interactions. This heralds a significant advancement in the methodological toolkit available for enzymatic research. The convergence of data from various experimental platforms increases our confidence in the reproducibility and generalizability of our findings. This contributes significantly to the broader scientific dialogue on enzyme kinetics and its applications in biotechnology and pharmaceuticals.

The correlation supports the claim that electrochemical impedance spectroscopy is an effective tool for understanding the fundamental electrochemical properties of the enterokinase enzyme. Electrochemical methods are advantageous due to their speed, precision, and cost-effectiveness, highlighting their potential usefulness. These techniques have the potential to change the way we assess basic biochemical properties of enzymes due to their unique attributes. This is applicable to both simple samples and specialized fusion proteins, where traditional techniques such as UV-Vis and fluorescence spectroscopy may not be adequate. Electrochemical impedance spectroscopy is a versatile and essential tool for biochemical investigations due to its cost-effectiveness, rapid execution, and precision. It is the dawn of a possible new way in the characterization of some enzyme properties.

## 4. Conclusions

Our study introduced several mutations into the enteropeptidase light chain to improve its specificity and catalytic efficiency. The altered enzymes were characterized through fluorescence and electrochemical impedance spectroscopy (EIS), showing variations in kinetic parameters that aligned with our objectives. The results from EIS provided valuable insights into enzymatic activities in complex conditions, highlighting EIS’s potential as a powerful alternative for enzyme kinetics analysis, particularly in environments where traditional methods are unsuitable. Our findings contribute to the understanding of enteropeptidases’ biochemical properties and highlight the practical applicability and accuracy of EIS in broader enzymatic research. The consistency between the EIS results and those obtained through conventional methods reinforced the validity of EIS in studying enzyme kinetics, especially for enzymes operating in non-ideal conditions. This study supports the broader adoption of electrochemical techniques in biochemical analysis and paves the way for further investigations into the enzymatic activities of genetically modified proteins.

## Figures and Tables

**Figure 1 sensors-24-02643-f001:**
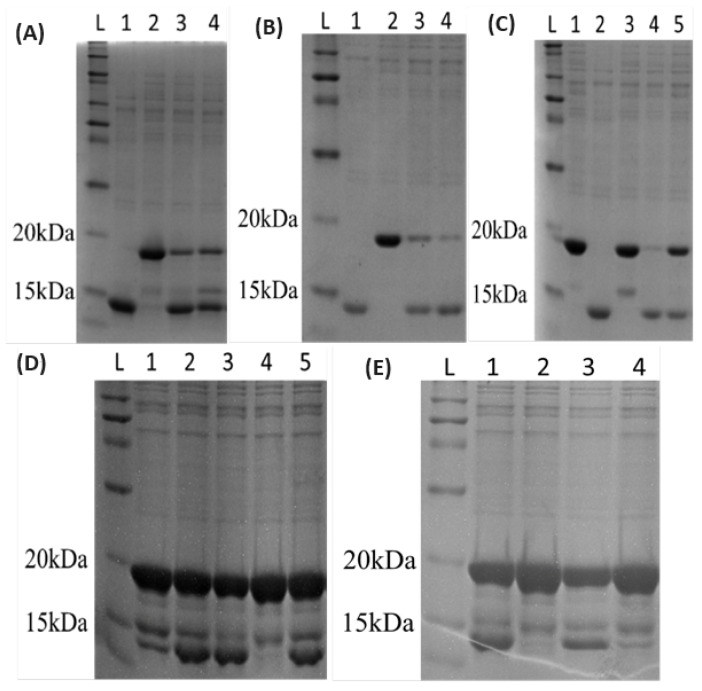
SDS-PAGE gels show the activity of the tested clones prepared from (**A**) Mut0—L ladder, 1 positive control, 2 negative control, 3,4 selected clones, (**B**) Mut1—L ladder, 1 positive control, 2 negative control, 3,4 selected clones, (**C**) Mut2—L—ladder, 1 negative control, 2 positive control. 3–5 selected clones, (**D**) Mut3—L ladder, 1 negative control, 2 positive control, 3–5 selected clones, (**E**) Mut4—L ladder, 1 positive control, 2 negative control, 3–4 selected clones. Substrate for cleavage was Trx-DCD1, which had a DDDDK amino acid residue sequence between thioredoxin and dermcidin. Protein ladder was Novex™ Sharp Pre-Stained Protein Standard (Invitrogen, Thermo Fisher Scientific, Waltham, MA, USA).

**Figure 2 sensors-24-02643-f002:**
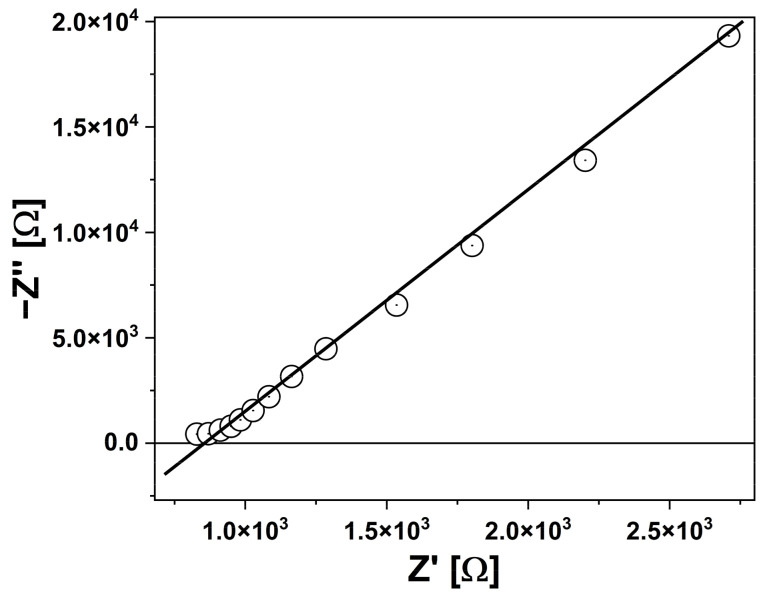
Nyquist diagram expressing the dependence of the imaginary component of the impedance −Z″ from its real component Z′ for Mut 4 with the substrate concentration 0.01 mM. The red line indicates the way of determining the electrolyte resistance parameter for a given enzyme and substrate concentration.

**Figure 3 sensors-24-02643-f003:**
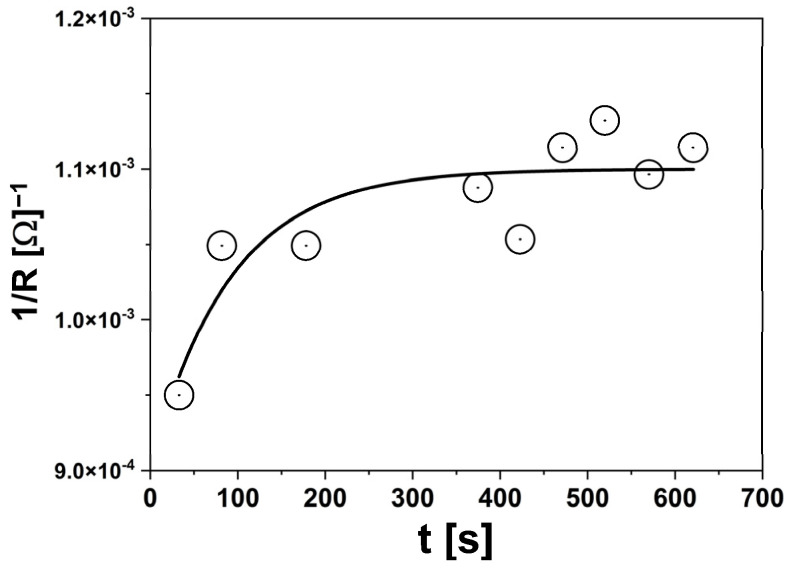
Change in system resistance detected by EIS for Mut3 mutant enzyme as a function of measurement time. The substrate concentration was 0.01 mM.

**Figure 4 sensors-24-02643-f004:**
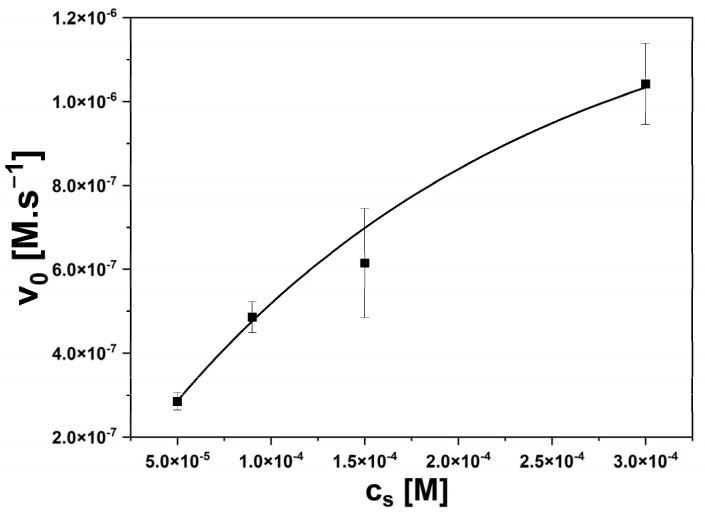
Graphical dependence of the Michaelis–Menten equation for the Mut1 enzyme with the substrate concentration 0.01 mM.

**Figure 5 sensors-24-02643-f005:**
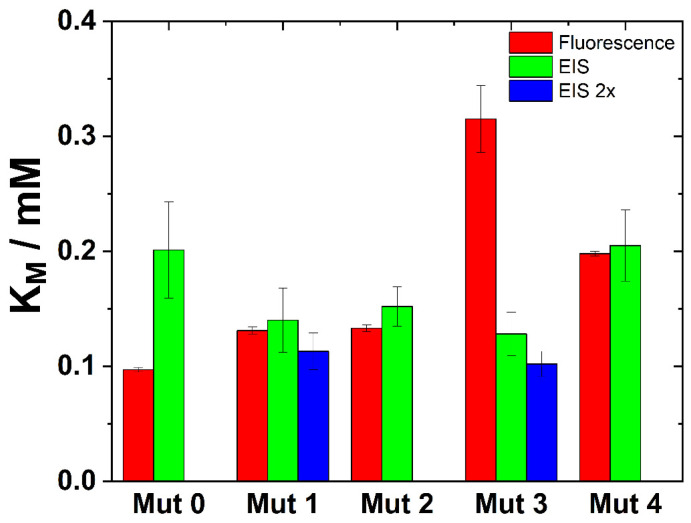
K_m_ values for the enzyme mutations studied and the two methods used to determine their kinetic parameters. EIS 2x denotes doubled enzyme concentrations.

**Table 1 sensors-24-02643-t001:** Specific activity of hEKL mutant forms.

Mutation	Change	Position	Amino Acid	Specific Activity (U∙mg^−1^)
Mut0	K99R	99	Lysin ⟶ arginine	7729
Mut1	N101S	101	Asparagine ⟶ serine	3615
Mut2	N95E	95	Asparagine ⟶ glutamic acid	6068
Mut3	N95D	95	Asparagine ⟶ aspartic acid	4453
Mut4	D100N	100	Aspartic acid ⟶ asparagine	4804

**Table 2 sensors-24-02643-t002:** Basic kinetic parameters of enzymes determined by fluorescence spectroscopy.

Enterokinase	Kinetic Parameters		
	K_m_ [∙mM]	k_cat_ [∙s^−1^]	k_cat_/K_m_ [∙mM^−1^∙s^−1^]
hEKL	0.116 ± 0.004	131.75 ± 20.660	1129.354
Mut0	0.097 ± 0.002	14.116 ± 0.716	144.535
Mut1	0.131 ± 0.003	54.050 ± 10.765	410.506
Mut2	0.133 ± 0.003	66.603 ± 8.208	498.275
Mut3	0.315 ± 0.029	15.098 ± 1.098	47.880
Mut4	0.198 ± 0.002	39.170 ± 2.269	197.828

One-sample *t*-test results show that by conventional criteria, the difference between engineered enzymes and hEKL was considered to be not statistically significant (*p* value equals 0.2030). Moreover, according to Grubbs’ test, although K_m_ for Mut3 was furthest from the rest, it was not a significant outlier (*p* > 0.05). This meant that the kinetic parameters of genetically engineered forms of enzymes did not improve significantly.

**Table 3 sensors-24-02643-t003:** Summary of all kinetic parameters for each enzyme mutation that were determined at basal substrate concentrations.

Enterokinase	Kinetic Parameters		
	K_m_ [∙mM]	k_cat_ [∙s^−1^]	k_cat_/K_m_ [∙mM^−1^∙s^−1^]
Mut0	0.206 ± 0.042	15.5 ± 2.0	75.2
Mut1	0.140 ± 0.028	75.9 ± 13.2	542.1
Mut2	0.152 ± 0.017	64.2 ± 11.1	422.3
Mut3	0.128 ± 0.019	21.9 ± 4.7	171.1
Mut4	0.205 ± 0.031	35.0 ± 12.4	170.7

**Table 4 sensors-24-02643-t004:** Summary of individual kinetic parameters for Mut1 and Mut3 that were determined at doubled enzyme concentrations.

Enterokinase	Kinetic Parameters		
	K_m_ [∙mM]	k_cat_ [∙s^−1^]	k_cat_/K_m_ [∙mM^−1^∙s^−1^]
Mut1	0.113 ± 0.016	66.6 ± 4.3	589.4
Mut3	0.102 ± 0.011	31.1 ± 5.4	304.9

## Data Availability

The data presented and analyzed in this study are available on reasonable request from the corresponding author.

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
