# Peer review of "Electrochemical Impedance Spectroscopy for the Sensing of the Kinetic Parameters of Engineered Enzymes"

_sensors, 2024, doi:10.3390/s24082643_

Round 1

Reviewer 1 Report

Comments and Suggestions for Authors

sensors-2958404 - (Title: Electrochemical impedance spectroscopy for the sensing of the kinetic parameters of engineered enzymes)

Comments

1.     The abstract session of thos manuscript is poorly presented. Authors should report a summary of results in the study that shows how they distinguish the performance of the two spectroscopy methods considered.

2.     Introduction session should be improved for instance the 2 paragraphs highlighted can be collapsed.”

Various investigations into the biochemical characteristics of human and bovine enteropeptidase have presented a complex and often contradictory landscape. Innovative methodologies are necessary to determine critical parameters such as the maximum reaction rate (vmax), Michaelis constant (Km), and turnover number (kcat) due to discrepancies in their values [6,20–25]. Given the challenges, our study uses EIS to investigate the biochemical properties of engineered enzymes. This method is known for its sensitivity and precisions in biological applications and offers new insights into enzyme kinetics analysis. [14].

The adoption of EIS in enzyme kinetics presents a transformative opportunity for biochemical research. Its versatility across different sample environments—ranging from clear to turbid and colored to colorless solutions—makes it a superior choice for varied research settings. Furthermore, the method's comparability in speed and accuracy with traditional fluorescence spectroscopy, alongside its reduced operational costs and increased user-friendliness, positions EIS as a significant advancement in enzymatic studies [14].

3.     Authors should show also the Nyquist plots for other mutations for the figures 2, 3 and 5

4.     EIS 2x in Figure 5 was not mentioned in any part of the manuscript only shown in figure 5, can authors define the originality of EIS 2x in figure 5

5.     Authors should use the statistical test to confirm the EIS and fluorescence method considered in this study e.g student t-test

Comments on the Quality of English Language

sensors-2958404 - (Title: Electrochemical impedance spectroscopy for the sensing of the kinetic parameters of engineered enzymes)

Comments

1.     The abstract session of thos manuscript is poorly presented. Authors should report a summary of results in the study that shows how they distinguish the performance of the two spectroscopy methods considered.

2.     Introduction session should be improved for instance the 2 paragraphs highlighted can be collapsed.”

Various investigations into the biochemical characteristics of human and bovine enteropeptidase have presented a complex and often contradictory landscape. Innovative methodologies are necessary to determine critical parameters such as the maximum reaction rate (vmax), Michaelis constant (Km), and turnover number (kcat) due to discrepancies in their values [6,20–25]. Given the challenges, our study uses EIS to investigate the biochemical properties of engineered enzymes. This method is known for its sensitivity and precisions in biological applications and offers new insights into enzyme kinetics analysis. [14].

The adoption of EIS in enzyme kinetics presents a transformative opportunity for biochemical research. Its versatility across different sample environments—ranging from clear to turbid and colored to colorless solutions—makes it a superior choice for varied research settings. Furthermore, the method's comparability in speed and accuracy with traditional fluorescence spectroscopy, alongside its reduced operational costs and increased user-friendliness, positions EIS as a significant advancement in enzymatic studies [14].

3.     Authors should show also the Nyquist plots for other mutations for the figures 2, 3 and 5

4.     EIS 2x in Figure 5 was not mentioned in any part of the manuscript only shown in figure 5, can authors define the originality of EIS 2x in figure 5

5.     Authors should use the statistical test to confirm the EIS and fluorescence method considered in this study e.g student t-test

Author Response

  1. The abstract session of thos manuscript is poorly presented. Authors should report a summary of results in the study that shows how they distinguish the performance of the two spectroscopy methods considered.

Thank you very much for this comment. We have improved abstract to be more compact and informative. In our study we wanted to improve the kinetic parameters of enteropeptidaze by changes of aminoacids at certain position its structure and, consequently, prove the suitability of EIS for the enzyme kinetics research. According to our results the kinetic parameters of engineered enzymes were not improve significantly as stated in the manuscript (added text bellow the Table 1). On the other side, the suitability of EIS was proved by our results, since the differences in the parameters obtained by both methods are not significant (according to t-test – added text to the manuscipt).

  1. Introduction session should be improved for instance the 2 paragraphs highlighted can be collapsed.”

Various investigations into the biochemical characteristics of human and bovine enteropeptidase have presented a complex and often contradictory landscape. Innovative methodologies are necessary to determine critical parameters such as the maximum reaction rate (vmax), Michaelis constant (Km), and turnover number (kcat) due to discrepancies in their values [6,20–25]. Given the challenges, our study uses EIS to investigate the biochemical properties of engineered enzymes. This method is known for its sensitivity and precisions in biological applications and offers new insights into enzyme kinetics analysis. [14].

The adoption of EIS in enzyme kinetics presents a transformative opportunity for biochemical research. Its versatility across different sample environments—ranging from clear to turbid and colored to colorless solutions—makes it a superior choice for varied research settings. Furthermore, the method's comparability in speed and accuracy with traditional fluorescence spectroscopy, alongside its reduced operational costs and increased user-friendliness, positions EIS as a significant advancement in enzymatic studies [14].

The introduction part of the manuscript was also rewritten.

  1. Authors should show also the Nyquist plots for other mutations for the figures 2, 3 and 5.

Thank you very much for this advice, but in our opinion, it would be too many very similar curves in one frame and the figure would become cluttered. The curves are very similar since the EIS measurements were done at OCP. Then only the capacitive behavior affected by the roughness of the electrode surface was observed (a pure capacitor cannot be used in the fitting, but a CPE must be used). Other differences in the shape of the Nyquist diagrams were not observed (nor could they be expected), so we do not think it is necessary to plot them. The curves in the aforementioned figures are only shown to make it clear to readers how the authors went about measuring, evaluating, and processing the results so they can use this method in their research.

  1. EIS 2x in Figure 5 was not mentioned in any part of the manuscript only shown in figure 5, can authors define the originality of EIS 2x in figure 5.

Thank you very much for this comment. This means that doubled enzyme concentration was used. We add this information to the figure caption.

  1. Authors should use the statistical test to confirm the EIS and fluorescence method considered in this study e.g student t-test

Reviewer is right, we have performed t-test and the comments are added to the manuscript.

Reviewer 2 Report

Comments and Suggestions for Authors

This article developed electrochemical impedance spectroscopy (EIS) to detect kinetic parameters of mutants of enteropeptidase hEKL. This study presented a pioneering approach to enzymatic kinetics using EIS to assess fundamental parameters of modified enteropeptidases. It was validated the use of EIS for measuring enzyme kinetics and provided insight into the effects of specific amino acid changes on enteropeptidase function. It was highlighted that EIS could provide a novel approach that overcomes some limitations. The authors make a great effort in this research. The research topic is within the scope of this journal. The experimental work appears somewhat technically sound, and the tables and figures convey the intended messages. This work is meaningful, and the novelty of this study is somewhat good. Before acceptance, this paper could be made a revision.

Q1. The section of “Materials and Methods” is not detailed. Which measurement is select to determine the concentration of all mutant forms of hEKL? Why does it show the concentration of all mutant forms of hEKL as 2.5 nM? What’s the pH of the solution? What’s the definition of hEKL activity? Should all mutants control the temperature of 37 oC? The authors should give the primers as SI.

Q2. What’s the method of Figure 1? Moreover, please show all the proteins of marker.

Q3. Figure 2, Figure 3, and Figure 4 can be shown as one figure.

Q4. What’s the amino acid sequence of hEKL? Additionally, please provide the gene sequence of hEKL as SI.

Q5. The authors claim that “selected these mutations based on high enzyme-substrate affinity or catalytic efficiency”. However, it is still unclear that how does it select the mutations K99R, N101S, N95E, N95D, and D100N. 

Q6. What are the culture media and culture conditions for the expression of hEKL by P. pastoris Y11430? Why does it choose P. pastoris Y11430 as the host for the protein expression of hEKL?

Comments on the Quality of English Language

no

Author Response

Please, see attached file.

Round 2

Reviewer 1 Report

Comments and Suggestions for Authors

Accept 

Reviewer 2 Report

Comments and Suggestions for Authors The authors have taken the recommendations suggested in the first review very seriously. I think this manuscript can now be published in Sensors.